# UNIMAX: FAIRER AND MORE EFFECTIVE LANGUAGE SAMPLING FOR LARGE-SCALE MULTILINGUAL PRE-TRAINING

**Hyung Won Chung**[*], **Noah Constant**[*], **Xavier Garcia**[*]
**Adam Roberts, Yi Tay, Sharan Narang, Orhan Firat**
Google Research
`h.w.chung27@gmail.com, {nconstant, xgarcia}@google.com`

## ABSTRACT

Pretrained multilingual large language models have typically used heuristic temperature-based sampling to balance between different languages. However previous work has not systematically evaluated the efficacy of different pretraining language distributions across model scales. In this paper, we propose a new sampling method, UNIMAX, that delivers more uniform coverage of head languages while mitigating overfitting on tail languages by explicitly capping the number of repeats over each language's corpus. We perform an extensive series of ablations testing a range of sampling strategies on a suite of multilingual benchmarks, while varying model scale. We find that UNIMAX outperforms standard temperature-based sampling, and the benefits persist as scale increases. As part of our contribution, we release: (i) an improved and refreshed mC4 multilingual corpus consisting of 29 trillion characters across 107 languages, and (ii) a suite of pretrained umT5 model checkpoints trained with UNIMAX sampling.

## 1 INTRODUCTION

State-of-the-art multilingual models (Xue et al., 2021; 2022; Goyal et al., 2021, *inter alia*) utilize large-scale self-supervised learning, which involves jointly training on many languages. Because data availability varies greatly across languages, multilingual pretraining can be characterized as multitask learning (or multi-objective optimization) with severe data imbalance. Typically English is the highest-resource language (or task) with orders of magnitude larger size than lower-resource languages. For example, in the mC4 corpus (Xue et al., 2021), English has roughly 9.7 trillion characters, which is over 92,000 times larger than the lowest resource language, Yoruba. As a result, a key problem in designing such models is the "language balancing" problem: in what proportions should we balance the pretraining languages? Deriving the optimal balance is a difficult open research problem due to the high cost of pretraining.

The standard approach to this problem has been to upsample languages with smaller datasets, using a temperature hyperparameter $\tau$ (Devlin et al., 2019). However, one shortcoming of this approach is that choosing $\tau$ based on the desired distribution among higher-resources languages may result in examples from the lowest-resource languages being repeated excessively. Figure 1a shows the number of epochs covered for each language in the mC4 corpus. When using $\tau = 3.33$ and a trillion token budget (the values used in popular models such as mT5 and ByT5), the lowest-resource languages are repeated over 100 times. This excessive repetition can have several unwanted consequences: (i) it leads to overfitting, which degrades performance on downstream tasks (Raffel et al., 2020; Lee et al., 2022; Hernandez et al., 2022), (ii) it increases the risk of memorizing private or sensitive content (Carlini et al., 2021; Lee et al., 2022), and (iii) it wastes training cycles that could have been devoted to unique examples. As models continue to grow in scale (Chowdhery et al., 2022; Brown et al., 2020; Smith et al., 2022), these issues with temperature sampling grow more pressing, as larger models benefit from longer training (Hoffmann et al., 2022), overfit more easily, and have a greater capacity to memorize.

---

[*]equal contribution

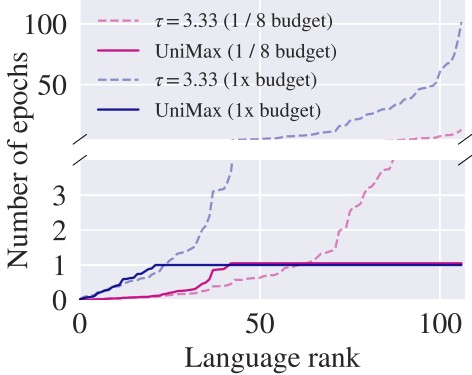 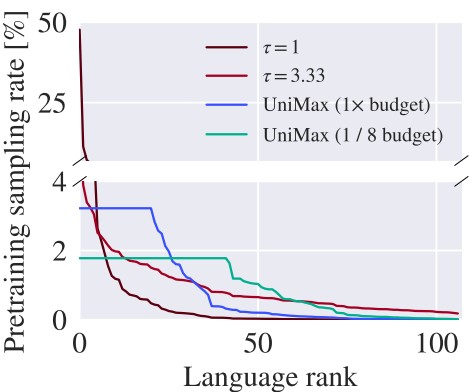

(a) Number of training epochs for each language. Temperature sampling results in a large number of data repeats for low-resource languages, whereas UNIMAX explicitly caps repeats.

(b) Pretraining sampling distribution. Temperature sampling results in poorly balanced distributions, whereas UNIMAX provides more uniform distributions without excessive upsampling.

Figure 1: The x-axis is the rank of the language based on the character count. $1/8$ budget refers to the 250,000 steps with sequence length of 512, which is one-eights of the full-scaling training budget (1M steps with 1024 sequence length) referred to as 1x, matching that of mT5.

This paper proposes a new paradigm for sampling across languages and datasets that ameliorates the above mentioned problems. We propose UNIMAX (uniform + max), a conceptually simple but highly effective two-pronged sampling approach that results in fairer and more effective language distributions for pretraining multilingual language models that work well across model scales. One of the main assumptions we make is that practical large-scale training jobs operate with a fixed amount of compute, which is often translated into a fixed training token budget (Raffel et al., 2020). UNIMAX starts by pre-allocating training tokens to underrepresented datasets based on the number of allowed **max** repeats ($N$). For the remaining budget, we prioritize "linguistic utility" (Blasi et al., 2022) by allocating **uniformly** across all languages with sufficient data to avoid exceeding the prescribed number of per-language epochs. Unlike previous approaches, this means UNIMAX is relatively resistant to distribution biases that arise due to artifacts of the corpus generation process (i.e., web crawlers). To take a concrete example, the mC4 corpus contains $70\times$ more English than Chinese text. While mT5's temperature sampling ($\tau = 3.33$) results in training on $3.4\times$ more English than Chinese, UNIMAX will assign equal training tokens to the two languages, provided that this doesn't result in repeating the 39 billion available Chinese tokens more than $N$ times.

Another key benefit of UNIMAX is that it is robust to model scaling. In considering language sampling strategies at scale, it is important to carefully control how many times a dataset can be repeated during training to avoid overfitting and memorization. Our proposed method explicitly controls the extent of data repeats of any language, providing a direct countermeasure to overfitting on low-resource languages, without imposing any reprioritization on higher-resource languages.

Our key contributions are to: (1) Propose UNIMAX, a simple but effective language sampling strategy that provides more uniform coverage of high-resource languages while mitigating overfitting on low-resource languages. (2) Perform an extensive series of ablations testing a range of sampling strategies on a suite of multilingual benchmarks, while varying model scale. (3) Release an improved and refreshed variant of the mC4 multilingual corpus consisting of 29 trillion characters across 107 languages. (4) Release pretrained model checkpoints using UNIMAX sampling.[1]

## 2 RELATED WORK

While (massively-)multilingual models enjoy the benefits of positive transfer across languages, the sheer number of languages reduces the effective capacity of the model per task. This competition

---

[1]https://github.com/google-research/t5x/blob/main/docs/models.md

among languages for limited model capacity is the well-known problem of "capacity bottleneck" (Arivazhagan et al., 2019), also known as the "curse of multilinguality" (Conneau et al., 2020).

In exploring the interaction between language balancing and model scale, previous work has ablated sampling temperature with models at or below 1 billion parameters (Conneau et al., 2020; Xue et al., 2021), but we believe our study is the first to systematically explore balancing strategies at scales above 1 billion parameters. Additionally, our work targets general-purpose encoder-decoder models, whereas most previous work used encoder-only models or targeted machine translation exclusively. Michel et al. (2021) explore adaptive mixing strategies in a similar setting, and also find that fixed uniform mixing is a strong baseline for multilingual pretraining.

Within the context of machine translation[2], Jean et al. (2019) proposed using a bi-level optimization algorithm maximizing the validation utility to adjust the sampling weights per language. Arivazhagan et al. (2019) was the first testing various temperatures ($\tau = 1, 5, 100$) to train a multilingual translation model for 212 language pairs. The study suggested over-sampling low-resource while sub-sampling high-resource languages to maximize a uniform utility. The over-training/over-fitting due to over-sampling was alleviated by increasing the number of tasks in that study. Wang et al. (2020b) proposed another bi-level approach, using differentiable data selection (Wang et al., 2020a). The study includes a uniform language distribution baseline ($\tau = \infty$), but this differs from UniMax in having no limit on repetition of examples in low-resource languages. Recently Wang et al. (2021) proposed using the gradient alignment information across languages to adjust the language weights, but it is unclear if this can be practically scaled to 100 or more languages.

Work on cross-lingual transfer learning has repeatedly found that English may not be the "ideal" source language for transfer. Malkin et al. (2022) conduct an analysis of zero-shot transfer, and calculate a "donation score" for each of 22 languages, measuring whether its inclusion in pretraining helps or hurts performance in other languages. They find English is among the "least donating"— one of only four languages whose inclusion results in an overall degradation in performance for other languages. In a similar exploration, Turc et al. (2021) find that German and Russian are more effective source languages for zero-shot transfer than English. These findings support the idea of testing sampling strategies where English doesn't outweigh other languages.

We focus here on the problem of balancing languages with significant web data. In particular, we limit our scope to the approximately hundred languages that occur with some frequency in the CommonCrawl corpus, as detected by CLD3 language detection. Other methods are likely needed to scale NLP methods to thousands of languages, e.g., see Wang et al. (2022) and Bapna et al. (2022).

## 3 SAMPLING METHODS

**Temperature-based sampling** In order to define a sampling distribution over languages, we use an empirical distribution for each language

$$p_l = \frac{n_l}{\sum_{l' \in L} n_{l'}} \tag{1}$$

where $n_l$ represents the "size" of language $l$, as discussed in Section §3.1. A temperature-based sampling strategy uses a distribution $q$ defined by exponentiating $p_l$ by the inverse of the temperature and renormalizing.

$$q_l = \frac{p_l^{1/\tau}}{\sum_{l' \in L} p_{l'}^{1/\tau}}. \tag{2}$$

Commonly used temperature values in the literature are summarized in Table 1. Higher temperature makes the distribution "flatter", approaching the uniform distribution as $\tau \to \infty$. In the other extreme with $\tau = 0$, the entire probability mass is contained in the language with the highest probability. Note, there is no guarantee that there exists a value of $\tau$ that achieves the desired balance on high- and mid-resource languages while still avoiding overfitting on the tail languages.

---

[2]We view the translation literature as directly relevant for language balancing research. However one key difference is that translation models used in balancing studies have assumed that English is always either the source or the target language. This may lead to preferring language distributions that cover more English.

Table 1: Language sampling temperatures of recent multilingual LLMs trained on unlabeled data.

|  | Temperature ($\tau$) |
| --- | --- |
| mBERT (Devlin et al., 2019) | 1.43 |
| XLM (Conneau & Lample, 2019) | 2.00 |
| XLM-R (Conneau et al., 2020) | 3.33 |
| mT5 (Xue et al., 2021) | 3.33 |
| XLM-E (Chi et al., 2022) | 1.43 |

**UNIMAX sampling**  For UNIMAX sampling, we start with a predefined character budget $C$. In practice, this is typically defined by the training compute allocated for the training job. The goal of UNIMAX sampling is to allocate the character budget to languages as uniformly as possible, without using more than $N$ epochs of any language. The first step is to sort the languages based on the character count in the training corpus. We iterate over languages starting from the one with the lowest character count. At each iteration, we check if the remaining character budget can be split evenly among the remaining languages without using more than $N$ epochs of any language. If so, we allocate the budget uniformly. If not, language $l$ is allocated $N$ epochs worth of characters and the remaining budget is reduced. Algorithm 1 formalizes this procedure.

---

**Algorithm 1: UNIMAX**

---

**Inputs :** Character count $c_l$ of each language $l$ in all the languages $L$ of the training corpus
Total character budget $C$
The number of epochs per language $N$
**Output:** Sampling distribution $p_l$ of each language
// Sort the languages by increasing number of character counts
$L \leftarrow \text{sortByCount}(L)$
$B \leftarrow C$ // Initialize the remaining budget to the total character budget
$i \leftarrow 0$
**for** $l$ *in* $L$ **do**
    $b_l \leftarrow \frac{B}{\text{len}(L)-i}$ // Compute the remaining budget per-language
    **if** $b_l > c_l \times N$ **then**
        // If per-language budget exceeds $N$ epochs of $l$, use $N$ epochs
        $U_l \leftarrow c_l \times N$
    **else**
        $U_l \leftarrow b_l$ // Otherwise use uniform per-language budget
    **end if**
    $B \leftarrow B - U_l$ // Update the remaining budget
    $i \leftarrow i + 1$
**end for**
$p \leftarrow \text{normalize}(U)$
**return** $p$

---

### 3.1 VOCABULARY GENERATION PROCESS

Following Xue et al. (2021), we use SentencePiece tokenization (Kudo & Richardson, 2018), which consumes a corpus to produce a subword-level tokenizer. This corpus is sampled from the available training data, typically using the same distribution used during model training. In the multilingual setting, however, this amounts to an instance of the language balancing problem. While many works explore varying the temperature of the *training* distribution, few perform the analogous change on the *vocabulary-learning* distribution. This could have drastic consequences in certain situations. For example, if the vocabulary learning distribution heavily favors Chinese, but the training distribution heavily favors English, models using this vocabulary may experience poor performance due to insufficient vocabulary coverage (Chung et al., 2020).

Most sampling strategies require some notion of "size" ($n_l$) of each language $l$. This is typically computed by counting words or tokens using a pre-existing multilingual tokenizer. But beyond the

unwanted complexity of requiring one "pre-tokenizer" to train a second tokenizer, such strategies do not generalize well to the massively multilingual setting, as tokenization across 100+ languages is non-trivial, particularly in languages written without spaces (e.g. Chinese, Japanese, Thai).

We propose to resolve the above issues by training separate vocabularies for each sampling strategy, using *character* counts as our measure of sub-corpus size. As previous work has tended to measure training budget in *tokens*, when comparing against such approaches, we assume a token contains 4 characters on average, following Xue et al. (2022). We use this same conversion rate to determine the total character budget $C$ in Algorithm 1, given a real training budget in terms of tokens.

# 4 EXPERIMENTS

## 4.1 PRETRAINING CORPUS

To ensure meaningful results when comparing language sampling strategies, it is critical that the examples in our pretraining corpus are *correctly* labeled for language. However language detection is far from a solved problem (Caswell et al., 2020), and audits have found low accuracy across a range of public multilingual datasets (Kreutzer et al., 2022). As one severe example, only 40% of the documents in the Marathi language bucket of the popular mC4 dataset (3.0.1) were found to be well-formed Marathi, with 50% coming from other languages, and 10% being non-language.

To mitigate this problem, we construct a new multilingual corpus by filtering mC4 (Xue et al., 2021) to remove documents whose language ID confidence is below 0.95. Compared to the original mC4 corpus (which used a threshold of 0.7), this removes 6.1% of documents; however only 5.1% of characters are removed, as the filtered low-confidence documents also tend to be shorter. The most filtered languages are Welsh (86%), Sindhi (85%) and Luxembourgish (84%). Inspection of several hundred examples across a subset of languages indicated a clear reduction in mislabeled documents.

## 4.2 VOCABULARY

As discussed in Section §3.1, we train a dedicated SentencePiece vocabulary for each language sampling method considered. We analyze these vocabularies in terms of token lengths and coverage of various writing scripts in Appendix A. Overall, the expected trends are observed that (i) higher temperature, (ii) smaller UNIMAX training budget, and (iii) higher UNIMAX max-epoch threshold ($N$) all lead to vocabularies that allocate capacity more uniformly across languages. This more uniform allocation results in fewer tokens from higher-resource language scripts (e.g. Latin), more tokens from lower-resource language scripts (e.g. Devanagari), and a shift towards shorter tokens.

## 4.3 EVALUATION TASKS

In selecting evaluation tasks, we aim to satisfy several key properties. First, tasks should be **linguistically diverse**, covering a range of languages from distinct families and regions, including both high- and low-resource languages. Second, tasks should be **free of language bias**. For example, the task training data and evaluation metrics should be well-balanced across languages. We also avoid benchmarks where English plays a special role, including datasets constructed in English and translated post-hoc to other languages, as well as zero-shot transfer tasks where English is the sole source language. Finally, to the degree possible, benchmarks should be **realistic**, such that performing better on the benchmark gives us confidence that a model will do better on actual tasks facing language technology users. This is in contrast to "intermediate structure" tasks such as part-of-speech tagging.

**TyDi QA** (Clark et al., 2020) is a multilingual question-answering benchmark covering a range of typologically diverse languages. Questions are written from scratch by native speakers in each language, ensuring culturally relevant content and the absence of "translationese". We use the "GoldP" task, which covers 9 languages. To evaluate candidate models, we use the "in-language multitask" setting (Hu et al., 2020)—fine-tuning on a mixture of all available languages, and evaluating in each language separately. To maximize per-language performance, we select per-language checkpoints based on the validation performance, and report validation metrics, as no test set is provided.

The **WMT21** shared task on large-scale multilingual machine translation (Wenzek et al., 2021) tests the ability of single model to translate across many languages. We focus on the "small track" tasks,

each testing translation between 6 languages, in all 30 combinations. As with TyDi QA, we fine-tune a single multilingual model on the mixture of all tasks and select per-language-pair checkpoints based on the test set performance.[3] To reduce heavy English bias, we restrict training data to 1,000 examples per language pair by randomly subsampling from the training data. Limiting the size of the fine-tuning dataset also increases the importance of transferring knowledge from pretraining, thereby making more apparent the differences between various pretraining sampling strategies.

We also evaluate on several widely adopted multilingual benchmarks: XNLI (Conneau et al., 2018), XQuAD (Artetxe et al., 2020), MLQA (Lewis et al., 2020) and PAWS-X (Yang et al., 2019). While these do not satisfy all of our desiderata above (e.g., some are translated from English and some skew towards high-resource languages), their popularity still makes them valuable points of reference.

## 4.4 TRAINING SETUP

We closely follow mT5 (Xue et al., 2021) for model architecture and training procedure. Specifically, we use an encoder-decoder Transformer architecture. We use the span corruption pretraining objective from T5 (Raffel et al., 2020) on a multilingual corpus consisting of 101 languages plus 6 Latin-script variants (e.g. `ru-Latn`). We use batch size of 1024 sequences where each sequence is defined by selecting a chunk of 568 tokens from the training corpus. This is then split into 512 input and 114 target tokens. For the results in Section §5, the number of training steps is 250,000 unless otherwise stated. Additional training details can be found in Appendix C.

## 5 RESULTS

### 5.1 PRETRAINING LOSS

One challenge in evaluating multilingual models is the relative paucity of benchmarks that cover a broad range languages and are free from language bias (see Section §4.3). Numerous studies have established strong correlation between pretraining and fine-tuning performance (Devlin et al., 2019; Narang et al., 2021; Tay et al., 2022, *inter alia*). Thus, in addition to targeted downstream evaluations, it can be valuable to monitor and analyze performance on held-out pretraining data, which by definition covers all pretraining languages. Figure 2 shows pretraining loss curves of models trained with three sampling strategies, for both English (`en`) and Yoruba (`yo`), the highest and lowest-resource languages in our study. We plot three model sizes, as model scale controls the tradeoffs between positive cross-lingual transfer and overfitting.

In Figure 2a, we observe that the loss values for `en` are much lower than for `yo`, and model scale alone does little to close this gap. This suggests that downstream performance on low-resource languages will suffer if they are not upsampled. On the other hand, too much upsampling is problematic, as shown in Figure 2b. Crucially, overfitting only emerges with scale: `Large` shows no obvious overfitting, `XL` shows weak overfitting, and it becomes conspicuous at `XXL` size. Moreover, while the effects of repetition may appear to be limited to Yoruba, results from Hernandez et al. (2022) suggest that overfitting in one language is likely to hurt performance across the board; the authors find that even repeating 0.1% of data 100 times can be as harmful as halving model size.

Figure 3 shows the pretraining loss curve for a longer train, corresponding to 1/2 character budget. In this case, overfitting emerges even for the `Large` model after around 300,000 steps. Given that previous work (Conneau et al., 2020; Xue et al., 2021) studied sampling distributions at smaller model sizes or for shorter training steps, the importance of overfitting may have been overlooked. Notably, across Figures 2c and 3c, UNIMAX closes the loss gap between high- and low-resource languages without showing any sign of overfitting.

### 5.2 DOWNSTREAM EVALUATIONS

Figure 4 shows the average TyDi QA performance for the three sampling strategies discussed in Section §3. Our first observation is that UNIMAX outperforms the other two consistently across

---

[3]We exclude results on language pairs with Serbian as the target language, due to unstable performance. The WMT21 Serbian training data covers Cyrillic and Latin scripts, and we observed high variance in eval metrics across checkpoints, depending on whether the model happened to use the same script as the references.

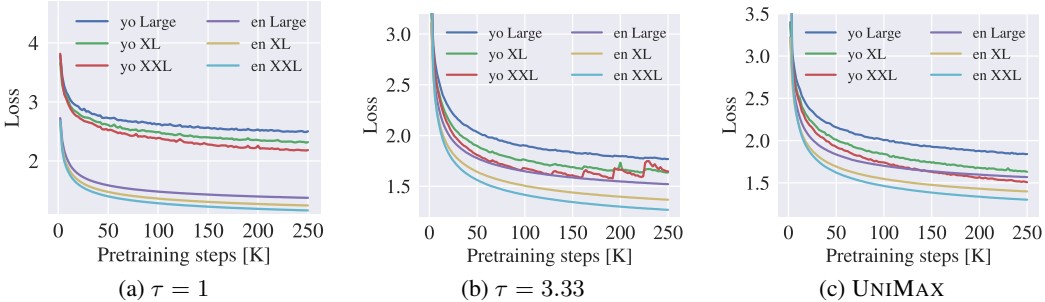

Figure 2: Pretraining cross-entropy loss on the held-out data over the training steps. With too-low temperature, low-resource languages are sampled too little, and their losses are relatively high. With higher temperature, overfitting becomes more severe with increasing model size. Note that loss values are not directly comparable across sampling strategies due to the difference in vocabulary.

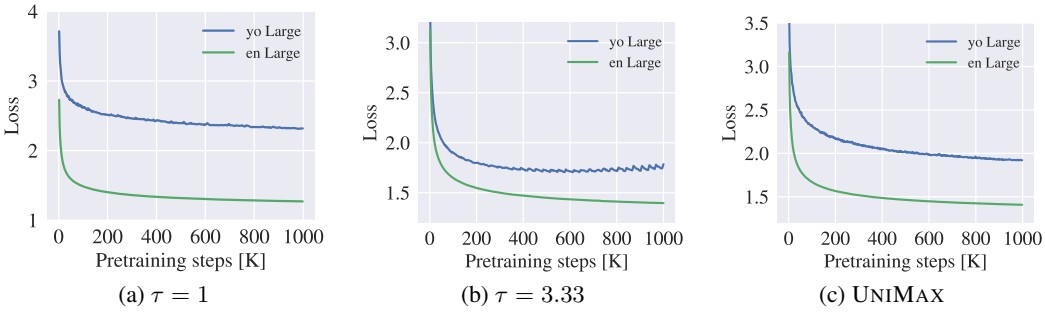

Figure 3: Pretraining cross-entropy loss on the held-out data over 1M training steps. With the sequence length of 512, this corresponds to $1/2$ character budget. The overfitting behavior emerges only after sufficient number of training steps for $\tau = 3.33$.

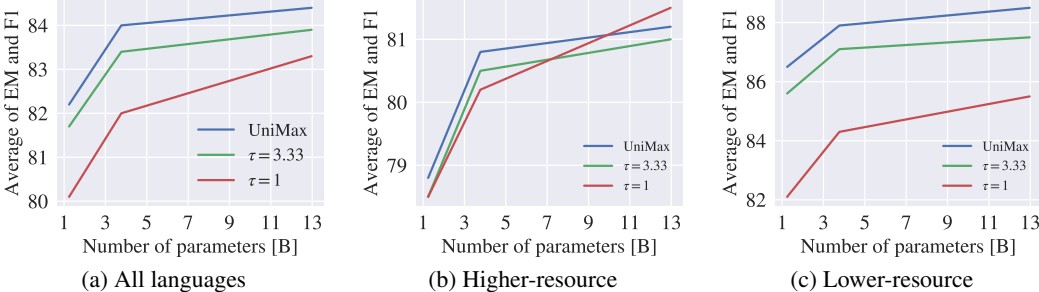

Figure 4: Average TyDi QA GoldP performance across three model sizes. Overall, UNIMAX outperforms both baselines at all model sizes considered. Breakdowns on higher-resource (top-5) and lower-resource (bottom-4) languages show UNIMAX outperforms $\tau = 3.33$ on both high- and low-resource, and only underperforms $\tau = 1$ on high-resource at large model scales.

model scales. We can get additional insights by correlating per-language performance with the pretraining portion of each language, as shown in Figure 5. Generally, languages seen more during pretraining tend to have higher performance, but this benefit seems to have its limit. For example, $\tau = 1$ sampling allocates 47.7% of training to en but this only translates to 1% better performance than UNIMAX, which allots only 1% to en. We believe this surprisingly small delta is attributable to the benefit of positive transfer from non-English languages onto English. Additionally, we note that UNIMAX outperforms $\tau = 3.33$ on Swahili, despite seeing fewer Swahili examples during training. See Table 7 for full per-language metrics across sampling strategies and model scales.

Figure 6 (left) shows that UNIMAX also outperforms temperature sampling on WMT21, across all model sizes considered. Figure 6 (right) shows that the benefit is spread out across the majority of language pairs, as opposed to only benefiting a small subset. We also note that the benefit is more

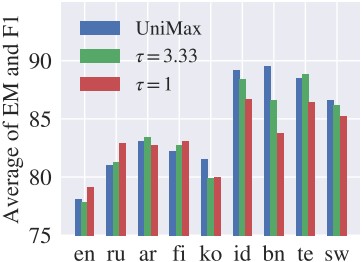 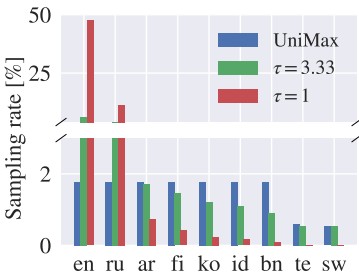

Figure 5: **Left**: Per-language TyDi QA performance of XXL models pretrained using three different sampling strategies. **Right**: Pretraining sampling rates of the three models.

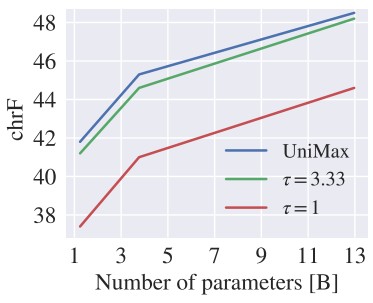 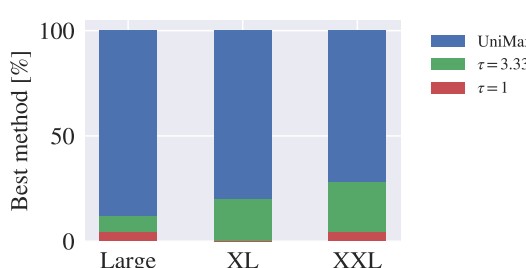

Figure 6: **Left**: WMT21 performance averaged across all language pairs. UNIMAX outperforms both baselines at all model sizes. **Right**: The majority of the language pairs benefits with UNIMAX.

pronounced for languages pairs where the target language is non-English, see Fig. 8 for details. Appendix D provides evaluations on additional benchmarks, which show similar trends.

## 5.3 FURTHER ABLATIONS

Our core experiments described in the previous sections were done across three model scales, but limited the character budget to $1/8$, due to compute resource constraints. In this section, we perform two additional ablations, but limiting to Large size.

First, in Table 2 (left) we show the effect of increasing the budget by $4\times$. Note, longer training changes the "shape" of the UNIMAX distribution (cf. Figure 1b), as the increased budget means more languages will hit their max-$N$ epoch cap. Overall, we find that UNIMAX still outperforms temperature sampling, with the longer train boosting performance across the board.

Second, we ablate the UNIMAX max-epoch parameter $N$. So far, we have used $N = 1$, i.e. no example is repeated. Table 2 (right) shows TyDi QA results for $N \in \{1, 5, 10\}$. We observe the best performance when disallowing repeats entirely, but the effect is small. We also note that the optimal setting of $N$ likely depends on the character budget, as for large enough budgets, UNIMAX-1 will see vanishingly little of the lowest-resource languages.

Table 2: Average TyDi QA results of additional Large-size models. **Left**: Comparison of sampling strategies at larger ($1/2$) character budget. **Right**: Ablation of UNIMAX max-epochs ($N$) parameter.

| | TyDi QA | | | TyDi QA |
|---|---|---|---|---|
| $\tau = 1.0$ | 81.2 | | $N = 1$ | **82.2** |
| $\tau = 3.33$ | 82.8 | | $N = 5$ | 81.5 |
| UNIMAX | **83.1** | | $N = 10$ | 81.8 |

Table 3: Comparison to mT5. XNLI and PAWS-X show average per-language accuracy; the rest show average per-language EM/F1. We use the translate-train setting except for TyDi QA, which uses "in-language". We omit results for the `Large` configuration due to instabilities in training.

| Model | XNLI | | PAWS-X | | XQuAD | | MLQA | | TyDi QA | |
|---|---|---|---|---|---|---|---|---|---|---|
| | mT5 | umT5 | mT5 | umT5 | mT5 | umT5 | mT5 | umT5 | mT5 | umT5 |
| Small | 72.0 | **76.2** | 79.9 | **87.2** | 49.4 / 64.5 | **60.5 / 74.0** | 38.8 / 56.6 | **41.8 / 60.7** | **62.7 / 74.0** | 56.6 / 70.0 |
| Base | 79.8 | **80.8** | 89.3 | **90.4** | 59.7 / 75.3 | **67.3 / 79.8** | 48.5 / 67.6 | **51.6 / 70.5** | 68.4 / 79.7 | **68.4 / 81.0** |
| XL | 85.3 | **86.5** | **91.0** | 90.7 | 56.6 / 75.1 | **75.0 / 86.1** | 54.5 / 73.5 | **58.3 / 76.8** | **78.4 / 87.6** | 74.1 / 85.2 |
| XXL | 87.1 | **87.8** | **91.5** | 91.2 | 71.3 / 85.2 | **77.9 / 88.2** | 57.4 / 76.0 | **70.5 / 78.6** | 79.5 / 88.7 | **81.2 / 89.7** |

# 6 UMT5 MODELS

We put our above findings into practice by training a suite of "umT5" models over a trillion tokens. For these final models, we also update the training corpus and add analysis comparing to mT5, which is the most direct point of comparison. As large language models are increasingly used for knowledge-intensive tasks (Petroni et al., 2019; Roberts et al., 2020; Petroni et al., 2021), it is important that training corpora are up-to-date. Given that the mC4 corpus is over two years old, we update the corpus (update version 3.1.0) to cover crawled documents through August 2022.

Beyond adding fresh documents, we make three changes to mC4. First, as in Section §4.1, we raise the language detection confidence threshold from 0.7 to 0.95, which we found to increase accuracy and reduce documents with little or no natural language. Second, we adjust the mC4 bad word filters to be *soft* filters, allowing a random 0.1% of documents with bad words to pass through. This ensures that models trained on our corpus will have at least a minimal exposure to any term, which is likely to help on tasks like toxicity detection. Finally, we remove from mC4's bad words lists any term that results in filtering >10% of documents from the language in question. Applying these changes, our resulting training corpus consists of 28.8 trillion characters from 9.0 billion documents—a 35% increase in documents over mC4. This increase is due primarily to added documents, and persists despite the more aggressive filtering during language detection. See Table 8 for full corpus statistics.

We closely follow the training setup and evaluation tasks from the mT5 paper for a fair comparison. Table 3 shows that umT5 outperforms mT5 on most tasks, across all sizes, and particularly at the largest size.[4] See Appendix E for an additional ablation isolating the effect of the data refresh.

# 7 CONCLUSION

In this paper, we introduced UNIMAX, a language sampling strategy that comes close to being uniform across languages—as close as possible without introducing harmful repetition. We showed this method performs well across several benchmarks and model scales, up to 13 billion parameters. Given the guarantees it provides against repetition, we expect UNIMAX is also well-suited to even larger model sizes, where issues of overfitting and memorization are known to grow more severe.

Our study focused on a single pretraining paradigm. Future work is needed to test if UNIMAX confers gains in other settings, such as: (i) encoder-only or decoder-only models, (ii) models trained with parallel data, and (iii) models with dedicated parameters per language (Pfeiffer et al., 2022).

Beyond evaluation metrics, there is an *a priori* reason to prefer more uniform language distributions: they can be seen as more equitable, in that they come closer to treating each language as an equally priority. However there is an important remaining question about whether "language" is the right unit over which to normalize. One alternative that we think deserves further exploration is inspired by the notion of "demographic utility" discussed by Blasi et al. (2022), which treats each *speaker* as equal in balancing across languages. See Appendix B for further analysis.

We are eager to see if future research can formulate successful sampling strategies that take into account *both* linguistic and demographic utility. Ideally the language distributions output by these strategies can lead to stronger-performing and more equitable pretrained models that better serve a wider range of users and use cases.

---

[4]We exclude Large size, as umT5-Large exhibited pretraining instability and underperformed Base on all metrics. We have retrained umT5-Large with a new random seed and will release the final checkpoints.

ACKNOWLEDGMENTS

We are grateful to Yuan Cao, Noah Fiedel, Ben Hutchinson, Kathy Meier-Hellstern, Slav Petrov, Sebastian Ruder and Siamak Shakeri for helpful comments and discussion.

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

## A  VOCABULARY ANALYSIS

We analyze the makeup of the sub-word vocabularies resulting from different language sampling strategies in Table 4. To see how capacity is allocated across languages, we measure the proportion

Table 4: Frequency (%) of tokens satisfying various conditions across vocabularies generated using different sampling strategies, as well as the mT5 vocabulary.

|  | $\tau = 1$ | $\tau = 3.33$ | UM1 4x | UM10 4x | UM1 1x | UM10 1x | mT5 |
|---|---|---|---|---|---|---|---|
| Latin | 67.0 | 54.0 | 56.3 | 47.9 | 50.2 | 49.5 | 46.6 |
| Cyrillic (e.g. Russian) | 17.7 | 13.0 | 12.0 | 14.6 | 14.1 | 12.6 | 10.7 |
| Han (Chinese/Japanese) | 3.2 | 4.3 | 4.9 | 4.2 | 4.5 | 4.0 | 7.2 |
| Arabic | 2.8 | 4.6 | 5.6 | 4.2 | 4.5 | 4.6 | 3.0 |
| Greek | 1.3 | 1.9 | 3.1 | 1.5 | 2.0 | 1.3 | 2.1 |
| Hangul (Korean) | 0.8 | 1.5 | 2.1 | 1.5 | 1.8 | 1.4 | 1.7 |
| Devanagari (e.g. Hindi) | 0.5 | 2.2 | 1.9 | 3.0 | 2.9 | 2.6 | 1.3 |
| Thai | 0.3 | 0.9 | 1.2 | 1.0 | 1.3 | 0.9 | 1.8 |
| Contains whitespace (_) | 56.1 | 51.8 | 52.8 | 51.5 | 52.3 | 50.9 | 22.5 |
| Contains punctuation | 0.4 | 0.4 | 0.4 | 0.5 | 0.4 | 0.5 | 4.2 |
| 1-char | 6.1 | 8.2 | 8.6 | 8.3 | 8.6 | 8.2 | 7.8 |
| 2-char | 4.3 | 6.6 | 6.2 | 6.5 | 6.3 | 6.7 | 10.1 |
| 3-char | 8.9 | 12.6 | 11.7 | 13.1 | 12.6 | 13.4 | 17.8 |
| 4-char | 12.4 | 15.4 | 14.3 | 16.0 | 15.4 | 16.3 | 21.8 |
| 5-char | 13.1 | 15.2 | 14.3 | 15.5 | 15.1 | 15.9 | 17.2 |
| 6-char | 12.1 | 12.8 | 12.6 | 12.9 | 12.8 | 13.1 | 10.5 |
| 7-char | 10.6 | 9.6 | 9.9 | 9.6 | 9.7 | 9.4 | 6.0 |
| 8-char | 9.1 | 7.0 | 7.6 | 6.7 | 7.0 | 6.5 | 3.8 |
| 9-char | 7.4 | 5.0 | 5.7 | 4.8 | 5.1 | 4.6 | 2.2 |
| 10+ char | 15.9 | 7.5 | 9.2 | 6.7 | 7.5 | 5.9 | 2.7 |

of tokens using scripts associated with specific languages or language groups. We observe that increasing $\tau$ from 1 to 3.33 reduces the amount of Latin-script and Cyrillic tokens, presumably due to heavy reduction of English and Russian, the two most prevalent languages in mC4. However, as expected, raising temperature increases allocation to lower-ranking languages (Arabic, Chinese, Greek, Hindi, Japanese, Korean, Thai). Overall, the UNIMAX vocabularies are fairly similar to $\tau = 3.33$, with the expected trend that increasing pretraining budget ($1\times \rightarrow 4\times$) shifts more allocation to high-resource languages, while increasing the max-epoch threshold (UM1 $\rightarrow$ UM10) shifts more allocation to low-resource languages.

Looking at token lengths, we observe three factors contribute to shorter tokens: higher temperature, smaller UNIMAX budget, and higher UNIMAX max-epoch threshold. This is expected, as if a few high-resource languages dominate the vocabulary training corpus, the sub-word optimizer will assign more capacity to rare words from these languages, as opposed to covering more frequent (and shorter) words from a wider range of languages.

Comparing to mT5, we note that our vocabularies have longer tokens overall, and include more tokens containing the SentencePiece meta-symbol indicating whitespace (_). We suspect these differences stem from mT5 using a whitespace-splitting "pre-tokenizer" to derive preliminary word counts which are fed to the sub-word training algorithm. By comparison, our vocabulary is trained on raw text, with no need for a pre-tokenizer. We also note that mT5 contains many more tokens including punctuation. This is most likely due to the presence of low-quality non-language documents that are filtered by our higher language detection threshold, as described in section §4.1.

# B   REPRESENTATION RATIOS

To shed light on how moving towards a more population-based sampling strategy would reshape language distributions, we define the "representation ratio" $R(l, t)$ of a language $l$ within a training distribution $t$ to be the ratio between the language's rate of use within $t$ and its rate of native speakers ($s$) among the world population ($w$), as shown in equation (3). We say a language is "overrepresented" ($R > 1$) or "underrepresented" ($R < 1$) to indicate that its prevalence in training is more or less than expected based on its global L1 speaker population.

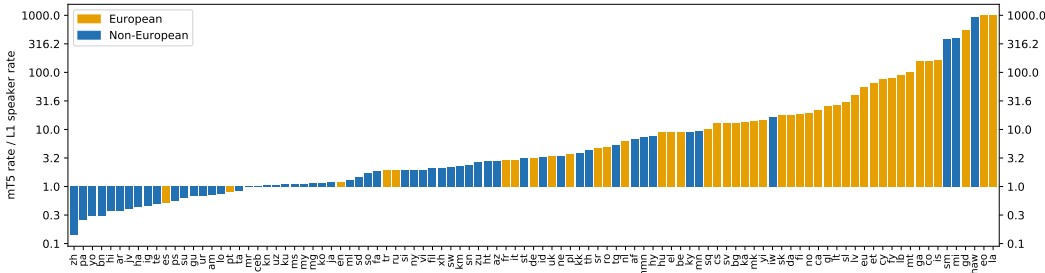

Figure 7: Representation ratios (mT5 pretraining rate / L1 speaker rate) of mT5 training languages. Esperanto (`eo`) and Latin (`la`) are clipped to 1,000; with few or no native speakers, the actual values are much higher.

$$R(l, t) = \frac{t_l / \sum_{l' \in L} t_{l'}}{s_l / w} = \frac{w \, t_l}{s_l \sum_{l' \in L} t_{l'}} \tag{3}$$

Figure 7 plots representation ratios for the mT5 language distribution, with native speaker counts taken from Wikipedia.[5] We observe a wide range of representation, ranging from $7\times$ underrepresented to over $900\times$ overrepresented. We also note a clear pattern whereby languages of Europe tend to have higher representation (Mann–Whitney $U=566$, $p<$1e$-5$, two-tailed), echoing a general bias observed by Blasi et al. (2022) for NLP resources.

Overall, we find that languages with large speaker populations, and particularly non-European languages are heavily underrepresented in commonly used pretraining distributions. For example, the top 5 underrepresented languages in mT5 training are all languages of Asia and Africa with 50 million or more native speakers: Chinese, Punjabi, Yoruba, Bengali and Hindi.

## C  ADDITIONAL TRAINING DETAILS

The model architectures used in this study are the same as mT5 models, except that relative position embeddings are not shared across layers. In all of our models, the vocabulary size is 256,000 subwords, and byte-level fallback is enabled, so unknown tokens are broken down into UTF-8 bytes.

We use the T5X library (Roberts et al., 2022) to train the models using Google Cloud TPUs. For pretraining, we use Adafactor optimizer (Shazeer & Stern, 2018) with a constant learning rate of 0.01 in the first 10,000 steps and inverse square root decay afterwards. For finetuning, we use Adafactor with a constant learning rate of 5e$-5$. Unlike mT5, we do not use loss normalization factor. Instead we use the number of real target tokens as the effective loss normalization.

Finally, we do not factorize the second moment of the Adafactor states and we also use momentum, neither of which are used in T5 and mT5 studies.

## D  ADDITIONAL BENCHMARKS

As a further comparison, we fine-tune and evaluate the nine pretrained models from Sections 4 and 5 on three additional benchmarks: XNLI zero-shot, XNLI translate-train, and XQuAD. We observe UNIMAX performs the best overall, although $\tau = 3.33$ is a close second.

## E  ABLATION ON mC4 REFRESH

To isolate the effect of refreshing the mC4 data, we train two additional models to 100,000 steps using UNIMAX sampling, under the same "full-budget" setting as in Section §6. These models differ only in that one trains on the original mC4 data, while the other uses our refreshed mC4 corpus.

---

[5]Languages not covered by mT5 are not plotted, but have a representation ratio of zero. As one example, Odia is a language of India spoken by 33 million native speakers, but not included in mT5 training.

Table 5: Additional benchmark results across sampling strategy and model scale. XNLI scores are average per-language accuracy. XQuAD is run in the translate-train setting and scores are average per-language EM/F1.

| | XNLI zero-shot | XNLI translate-train | XQuAD |
|---|---|---|---|
| *Large (1.2B)* | | | |
| $\tau = 1.0$ | 74.8 | 82.0 | 71.0/82.6 |
| $\tau = 3.33$ | 78.2 | 82.3 | **71.6/83.2** |
| UNIMAX | **78.3** | **82.7** | **71.6**/83.1 |
| *XL (3.7B)* | | | |
| $\tau = 1.0$ | - | 84.1 | 73.8/84.9 |
| $\tau = 3.33$ | - | **85.0** | 74.2/85.3 |
| UNIMAX | - | **85.0** | **74.5/85.5** |
| *XXL (13B)* | | | |
| $\tau = 1.0$ | - | 85.1 | 75.1/86.0 |
| $\tau = 3.33$ | - | **85.5** | 75.4/86.3 |
| UNIMAX | - | **85.5** | **75.6/86.4** |

Table 6: Ablation on mC4 refresh. XNLI scores are average per-language accuracy in the translate-train setting.

| | XNLI |
|---|---|
| mC4 | 77.6 |
| Refreshed mC4 | 77.7 |

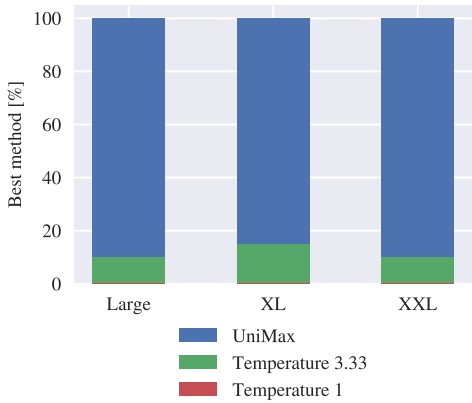

Figure 8: WMT21 results restricting to language pairs with non-English target. UNIMAX performs best on the vast majority of pairs, across all scales.

Table 7: Per-language TyDi QA GoldP performance, as average of exact-match and F1 metrics. Numbers in brackets represent the rank of the language in the pretraining corpus (e.g. `ru` has the second largest character count).

|  | Avg | en [1] | ru [2] | ar [15] | fi [23] | ko [27] | id [33] | bn [37] | te [59] | sw [62] |
|---|---|---|---|---|---|---|---|---|---|---|
| *Large (1.2B)* | | | | | | | | | | |
| $\tau = 1.0$ | 80.1 | **78.7** | 77.8 | 81.0 | 79.0 | 76.1 | 83.4 | 79.4 | 85.5 | 80.0 |
| $\tau = 3.33$ | 82.0 | 76.2 | 77.3 | **81.5** | 80.1 | 77.6 | **85.9** | 84.2 | 87.7 | 84.7 |
| UNIMAX | **83.3** | 76.5 | **78.6** | 80.8 | **80.4** | **77.9** | **85.9** | **86.5** | **87.9** | **85.6** |
| *XL (3.7B)* | | | | | | | | | | |
| $\tau = 1.0$ | 81.7 | 78.4 | 81.0 | 82.5 | 81.7 | 77.5 | **87.0** | 81.7 | 86.4 | 82.2 |
| $\tau = 3.33$ | 83.4 | **79.0** | 79.5 | **83.1** | 81.2 | 79.6 | 86.3 | 86.5 | 88.3 | 87.2 |
| UNIMAX | **83.9** | 78.0 | **81.3** | 82.3 | **81.9** | **80.4** | 86.8 | **88.9** | **88.4** | **87.7** |
| *XXL (13B)* | | | | | | | | | | |
| $\tau = 1.0$ | 82.2 | **79.1** | **82.9** | 82.7 | **83.1** | 80.0 | 86.7 | 83.8 | 86.4 | 85.2 |
| $\tau = 3.33$ | 84.0 | 77.8 | 81.3 | **83.4** | 82.7 | 79.9 | 88.4 | 86.6 | **88.8** | 86.2 |
| UNIMAX | **84.4** | 78.1 | 81.0 | 83.1 | 82.2 | **81.5** | **89.2** | **89.5** | 88.5 | **86.6** |

Table 6 shows results fine-tuning these two models on the XNLI task. We observe that the data refresh alone only gives a small boost (+0.1), supporting the view that the gains of UNIMAX over mT5 (+1.0) in Table 3 are primarily due to improved language sampling. While the inclusion of more recently crawled documents does not help on this particular benchmark (which is also several years old), we expect that the refreshed data will be useful to practitioners, and should help on tasks requiring up-to-date knowledge.

# F  ADDITIONAL TABLES

Table 8: Statistics for our improved and refreshed variant of the mC4 corpus, as well as the sampling rates (%) of the sampling methods studied in the paper.

| Lang | Chars (B) | $\tau = 3.33$ | $\tau = 1$ | UniMax (1×) | UniMax (1/8) | Lang | Chars (B) | $\tau = 3.33$ | $\tau = 1$ | UniMax (1×) | UniMax (1/8) |
|---|---|---|---|---|---|---|---|---|---|---|---|
| en | 13,396 | 5.75 | 46.58 | 3.22 | 1.48 | af | 7.4 | 0.61 | 0.03 | 0.16 | 1.27 |
| ru | 3,018 | 3.68 | 10.49 | 3.22 | 1.48 | be | 7.4 | 0.60 | 0.03 | 0.16 | 1.26 |
| es | 2,052 | 3.28 | 7.13 | 3.22 | 1.48 | kn | 6.9 | 0.59 | 0.02 | 0.15 | 1.18 |
| de | 1,799 | 3.15 | 6.26 | 3.22 | 1.48 | eu | 6.3 | 0.58 | 0.02 | 0.13 | 1.08 |
| fr | 1,433 | 2.94 | 4.98 | 3.22 | 1.48 | te | 5.9 | 0.57 | 0.02 | 0.13 | 1.01 |
| it | 779 | 2.45 | 2.71 | 3.22 | 1.48 | tg | 5.4 | 0.55 | 0.02 | 0.12 | 0.93 |
| pt | 658 | 2.33 | 2.29 | 3.22 | 1.48 | mt | 5.2 | 0.54 | 0.02 | 0.11 | 0.89 |
| zh | 556 | 2.21 | 1.93 | 3.22 | 1.48 | uz | 4.8 | 0.53 | 0.02 | 0.10 | 0.82 |
| pl | 527 | 2.18 | 1.83 | 3.22 | 1.48 | la | 4.5 | 0.52 | 0.02 | 0.10 | 0.78 |
| vi | 409 | 2.02 | 1.42 | 3.22 | 1.48 | so | 4.4 | 0.52 | 0.02 | 0.10 | 0.76 |
| nl | 371 | 1.96 | 1.29 | 3.22 | 1.48 | my | 4.2 | 0.51 | 0.01 | 0.09 | 0.72 |
| tr | 349 | 1.93 | 1.22 | 3.22 | 1.48 | sw | 4.1 | 0.51 | 0.01 | 0.09 | 0.70 |
| ar | 252 | 1.75 | 0.88 | 3.22 | 1.48 | ky | 3.7 | 0.49 | 0.01 | 0.08 | 0.64 |
| ro | 250 | 1.74 | 0.87 | 3.22 | 1.48 | gu | 3.6 | 0.49 | 0.01 | 0.08 | 0.61 |
| ja | 240 | 1.72 | 0.83 | 3.22 | 1.48 | km | 3.5 | 0.48 | 0.01 | 0.07 | 0.60 |
| cs | 236 | 1.71 | 0.82 | 3.22 | 1.48 | eo | 3.3 | 0.48 | 0.01 | 0.07 | 0.57 |
| fa | 202 | 1.63 | 0.70 | 3.22 | 1.48 | cy | 3.1 | 0.47 | 0.01 | 0.07 | 0.53 |
| sv | 201 | 1.63 | 0.70 | 3.22 | 1.48 | si | 3.0 | 0.46 | 0.01 | 0.06 | 0.52 |
| hu | 185 | 1.59 | 0.64 | 3.22 | 1.48 | ru-Latn | 2.6 | 0.44 | 0.01 | 0.06 | 0.44 |
| uk | 171 | 1.55 | 0.59 | 3.22 | 1.48 | pa | 2.2 | 0.42 | 0.01 | 0.05 | 0.37 |
| el | 166 | 1.54 | 0.58 | 3.22 | 1.48 | ga | 2.1 | 0.42 | 0.01 | 0.05 | 0.36 |
| da | 132 | 1.44 | 0.46 | 2.83 | 1.48 | zh-Latn | 1.9 | 0.40 | 0.01 | 0.04 | 0.33 |
| fi | 120 | 1.40 | 0.42 | 2.58 | 1.48 | ps | 1.4 | 0.37 | 0.01 | 0.03 | 0.25 |
| no | 116 | 1.38 | 0.40 | 2.49 | 1.48 | ku | 1.3 | 0.36 | 0.00 | 0.03 | 0.22 |
| bg | 99 | 1.32 | 0.35 | 2.13 | 1.48 | lb | 1.3 | 0.36 | 0.00 | 0.03 | 0.22 |
| th | 92 | 1.29 | 0.32 | 1.97 | 1.48 | ha | 1.1 | 0.34 | 0.00 | 0.02 | 0.19 |
| sk | 79 | 1.23 | 0.27 | 1.69 | 1.48 | ceb | 1.1 | 0.34 | 0.00 | 0.02 | 0.19 |
| hi | 75 | 1.21 | 0.26 | 1.60 | 1.48 | fy | 1.0 | 0.33 | 0.00 | 0.02 | 0.17 |
| ko | 74 | 1.21 | 0.26 | 1.59 | 1.48 | mg | 0.9 | 0.33 | 0.00 | 0.02 | 0.16 |
| lt | 63 | 1.15 | 0.22 | 1.35 | 1.48 | am | 0.9 | 0.32 | 0.00 | 0.02 | 0.16 |
| iw | 57 | 1.12 | 0.20 | 1.23 | 1.48 | el-Latn | 0.9 | 0.32 | 0.00 | 0.02 | 0.15 |
| ca | 55 | 1.11 | 0.19 | 1.19 | 1.48 | sd | 0.9 | 0.32 | 0.00 | 0.02 | 0.15 |
| id | 51 | 1.08 | 0.18 | 1.09 | 1.48 | gd | 0.8 | 0.31 | 0.00 | 0.02 | 0.14 |
| sl | 47 | 1.05 | 0.16 | 1.00 | 1.48 | ht | 0.8 | 0.31 | 0.00 | 0.02 | 0.14 |
| et | 42 | 1.02 | 0.15 | 0.91 | 1.48 | yi | 0.8 | 0.31 | 0.00 | 0.02 | 0.13 |
| lv | 38 | 0.99 | 0.13 | 0.83 | 1.48 | lo | 0.8 | 0.31 | 0.00 | 0.02 | 0.13 |
| bn | 34 | 0.95 | 0.12 | 0.72 | 1.48 | hi-Latn | 0.7 | 0.30 | 0.00 | 0.02 | 0.12 |
| sq | 18 | 0.79 | 0.06 | 0.39 | 1.48 | zu | 0.7 | 0.30 | 0.00 | 0.02 | 0.12 |
| az | 18 | 0.79 | 0.06 | 0.39 | 1.48 | jv | 0.7 | 0.30 | 0.00 | 0.01 | 0.12 |
| sr | 18 | 0.79 | 0.06 | 0.39 | 1.48 | hmn | 0.6 | 0.29 | 0.00 | 0.01 | 0.11 |
| ta | 17 | 0.78 | 0.06 | 0.37 | 1.48 | mi | 0.6 | 0.28 | 0.00 | 0.01 | 0.10 |
| ms | 15 | 0.75 | 0.05 | 0.32 | 1.48 | co | 0.5 | 0.27 | 0.00 | 0.01 | 0.09 |
| is | 14 | 0.73 | 0.05 | 0.30 | 1.48 | su | 0.5 | 0.27 | 0.00 | 0.01 | 0.09 |
| kk | 13 | 0.72 | 0.05 | 0.28 | 1.48 | ny | 0.5 | 0.27 | 0.00 | 0.01 | 0.08 |
| mr | 13 | 0.72 | 0.05 | 0.28 | 1.48 | xh | 0.5 | 0.27 | 0.00 | 0.01 | 0.08 |
| ne | 11 | 0.68 | 0.04 | 0.23 | 1.48 | st | 0.5 | 0.27 | 0.00 | 0.01 | 0.08 |
| ur | 11 | 0.68 | 0.04 | 0.23 | 1.48 | sm | 0.4 | 0.25 | 0.00 | 0.01 | 0.07 |
| ka | 10 | 0.67 | 0.04 | 0.22 | 1.48 | sn | 0.4 | 0.25 | 0.00 | 0.01 | 0.07 |
| hy | 10 | 0.66 | 0.03 | 0.21 | 1.48 | ig | 0.4 | 0.25 | 0.00 | 0.01 | 0.07 |
| mk | 10 | 0.65 | 0.03 | 0.20 | 1.48 | ja-Latn | 0.4 | 0.25 | 0.00 | 0.01 | 0.06 |
| fil | 9.5 | 0.65 | 0.03 | 0.20 | 1.48 | haw | 0.4 | 0.24 | 0.00 | 0.01 | 0.06 |
| ml | 9.4 | 0.65 | 0.03 | 0.20 | 1.48 | yo | 0.3 | 0.24 | 0.00 | 0.01 | 0.06 |
| mn | 9.3 | 0.65 | 0.03 | 0.20 | 1.48 | bg-Latn | 0.1 | 0.16 | 0.00 | 0.00 | 0.01 |
| gl | 8.8 | 0.64 | 0.03 | 0.19 | 1.48 | | | | | | |

