# OpenReview forum: "UniMax: Fairer and More Effective Language Sampling for Large-Scale Multilingual Pretraining"
_ICLR.cc/2023/Conference — ICLR 2023 poster_

### Official Review · Reviewer_JVbY · 2022-10-23

**Confidence:** 4
**Correctness:** 3
**Technical Novelty And Significance:** 3
**Empirical Novelty And Significance:** 3
**Recommendation:** 6

**Clarity, Quality, Novelty And Reproducibility:**

- The paper is well-written and extensive experiments have been systematically reported.
- The paper adds to the body of work on sampling methods for multilingual training. The sampling method provides some benefits in some downstream tasks.
- An additional artifact of the work is the availability of a larger and cleaner version of the mC4 corpus.

**Strength And Weaknesses:**

**Strengths**

- Extensive experimentation over large models and long pre-training provide strong evidence for the results.
- The UNIMAX method is simple to implement.
- The results show that close to uniform sampling works well and repetition may not very useful which is promising for language equity and efficient use of compute cycles.

**Weaknesses**

- The paper performs evaluation only on 2 tasks. Evaluation on a wider range of tasks would provide stronger evidence for the conclusions.
- The cross-lingual evaluations have been performed in in-language/translate-train settings. To understand the impact of the sampling strategy on cross-lingual representations, it would be better to report results in the cross-lingual zero-shot setting.

**Summary Of The Paper:**

In this paper, the authors explore an alternative method to temperature sampling for multilingual data sampling. The central premise is that existing methods can oversample low-resource languages resulting in overfitting and memorization. To alleviate this problem, they propose UNIMAX which caps the number of repetitions for low-resource languages while sampling high-resource languages uniformly. The experiments show that UNIMAX reduces overfitting for low-resource languages during pre-training and also improves performance on some downstream tasks.

**Summary Of The Review:**

The paper addresses the question of overfitting on low-resource languages while performing multilingual sampling. The results show close to uniform sampling holds promise, but evaluation on more tasks and settings as mentioned above is called for to make general claims.

---

> ### Author Response · Authors · 2022-11-19
> **Response to Reviewer JVbY**
>
> Re: only 2 eval tasks — We have **added experiments on XNLI zero-shot, XNLI translate-train, XQuAD and PAWS-X**, in §6 and Appendix D.
>
> Re: better to report results in cross-lingual zero-shot setting — We have **added results on zero-shot XNLI** in Appendix D.

---

### Official Review · Reviewer_JYyD · 2022-10-24

**Confidence:** 4
**Correctness:** 3
**Technical Novelty And Significance:** 2
**Empirical Novelty And Significance:** 3
**Recommendation:** 3

**Clarity, Quality, Novelty And Reproducibility:**

The paper is easy to follow. The novelty is limited. Data and codes will be released for reproduction.

**Strength And Weaknesses:**

Strengths:

The proposed model is simple yet effective in experiments. Large-scale experiments on mT5 model and mC4 data are conducted to show the effectiveness. Various downstream tasks are verified.


Weaknesses:

The claim that the proposed UNIMAX outperforms the temperature-based sampling is not well supported by experiments. According to Section 5, the UNIMAX outperforms on low-resource languages while underperforms on high-resource languages, although the average performance over multiple languages is improved. Prior works (even the temperature-based sampling itself) already found that larger improvments on low-resource languages with smaller degradation on high-resource languages would result in better average performance. A method which for example improves on low-resource languages and keeps the performance on high-resource languages is expected.

Experiments are not convincing enough. Limited evaluation tasks (TyDi QA and Multilingual NMT) are selected without good reasons to ignore others. The comparison to mT5 is unfair given different data and settings. In addition, it is unknown whether the proposed method would improve on other pretrained language models like mBERT. And could it be used on parallel data for training like XLM, mBART?


**Summary Of The Paper:**

This paper proposes a new sampling method to solve the language-imbalance problem in multilingual pretraining. The new method generally upsamples low-resource languages to a fixed epochs and uniformly sample high-resource languages. Experiments are conducted following mT5 and show the proposed method is better than the widely-used temperature-based sampling.

**Summary Of The Review:**

See the section of Strength and Weaknesses

---

> ### Author Response · Authors · 2022-11-19
> **Response to Reviewer JYyD**
>
> Re: outperforming on low-resource languages — Please note that with a max-1 epoch cap, UniMax actually *undersamples* low-resource languages compared to temp=3.33, as shown in Fig 1b. We found it remarkable that UniMax can outperform temp=3.33 on a low-resource language like Swahili (see Fig 5) despite seeing less pretraining data in Swahili. **We have adjusted the text** in §5.2 to better highlight this finding.
>
> Re: evaluation tasks are selected without good reasons to ignore others — The first paragraph of §4.3 lists our desiderata for evaluation tasks. Many standard multilingual tasks do not satisfy these criteria. However, we agree that reporting results on standard evals is important, so have **added experiments on XNLI zero-shot, XNLI translate-train, XQuAD, and PAWS-X**.
>
> Re: limited evaluation tasks — Beyond the new evaluations (see above), we also note that the pretraining loss on held-out training data (§5.1) can serve as a meaningful proxy evaluation metric, as many studies have established strong correlation between pretraining and fine-tuning performance.
>
> Re: comparison to mT5 is unfair — We agree the comparison in §6 is not apples-to-apples, as multiple factors differ (sampling strategy, pretraining data). We have **added an ablation by training two additional “full-budget” models** on the original mC4 data to isolate and measure the separate effects of pretraining data and sampling strategy. See Appendix E.
>
> Re: other pretrained language models like mBERT — We agree that testing other models would be interesting, but we have limited resources for additional pretraining experiments. We prioritized training encoder-decoder models as this architecture achieves state-of-the-art for many tasks, and is more flexible than encoder-only models like mBERT, as it extends to both understanding and generation tasks.
>
> Re: Could it be used on parallel data for training like XLM, mBART? — Thanks, yes, an approach similar to UniMax could definitely be used to balance parallel data. Extra consideration would be needed to simultaneously balance across source and target languages. Note, most parallel datasets assume English as the sole “bridge” language (always used as either source or target), including the datasets used in XLM pretraining and mBART fine-tuning. The heavy English bias would make it harder to interpret language sampling experiments in this context. Additionally, Kale et al. 2021 (https://aclanthology.org/2021.acl-short.87) finds that the value of parallel data in pretraining decreases with model size. For these reasons we opted to limit our scope to monolingual (non-parallel) pretraining data.

---

### Official Review · Reviewer_9yE2 · 2022-10-24

**Confidence:** 4
**Correctness:** 3
**Technical Novelty And Significance:** 2
**Empirical Novelty And Significance:** 3
**Recommendation:** 5

**Clarity, Quality, Novelty And Reproducibility:**

**Clarity**
- Overall, the paper is well-written. The motivation is clear.
- Technical issue with the Algorithm, particularly on the normalization part. $p_l$ <- Normalize($U_l$) should be $p$ <- Normalize($U$) instead?

**Quality**
- The paper quality meets the expectation of a good submission. No major typograpical errors.

**Novelty**
- In terms of technical novelty, this approach is not essentially new.
- The paper focuses on analysis and empirical results. The study can be helpful in training a model with a vast number of languages.

**Reproducibility**
- It is possible to reproduce the results with some difficulties (i.e., some hyperparameters are not detailed mentioned). It would be great to release the code and dataset to help other researchers to reproduce the results.

**Strength And Weaknesses:**

*Strengths:*
- A simple yet effective method. The approach is straightforward, and the results outperform the other sampling ratio baselines.
- The authors plan to release the datasets. It is a plus for reproducibility.

*Weaknesses:*
- The novelty is limited.
- Lack of analysis on how the sampling method affects the low-resource languages. I would suggest adding a specific section to describe the effects of sampling on low-resource languages. It would be great if the authors could also group the languages on the downstream tasks, so it is easier for the reader to get insights.
- There is a technical issue on the Algorithm 1. $p_l$ <- Normalize($U_l$) should be $p$ <- Normalize($U$) instead?

**Summary Of The Paper:**

The paper proposes UniMax, a simple language data sampling strategy to provide a more uniform coverage of high-resource languages. This allows the training to be more resistant to overfitting and memorization on low-resource languages since they are repeated excessively on a standard upsampling strategy. The paper also performs an extensive experiments on a series of datasets on various multilingual benchmarks. The reported findings are useful for practitioners and researchers who are working on multilingual large language models.

**Methods**
- The proposed method is relatively simple and straightforward.

**Efficiency**
- The paper also highlights the effective usage of computing budget, which is very important in building very large models.



**Summary Of The Review:**

In general, the paper provides interesting idea to investigate the sampling ratio to effectively address the overfitting issue on languages with much less data, and, it is practically useful. Indeed, a good empirical paper; however, the novelty of the paper is very limited, and I think the analysis of how changing the ratio affects the training loss and final performance is one of the important contributions of this paper.

For now, I would give "marginally below the acceptance threshold". And I would be happy to adjust my score if the authors address the concerns mentioned above.

---

> ### Author Response · Authors · 2022-11-19
> **Response to Reviewer 9yE2**
>
> Re: novelty — Please see our overall response.
>
> Re: lack of analysis on how the sampling method affects the low-resource languages — We found that many popular multilingual benchmarks do not adequately represent low-resource languages. We attempted to address this issue in §5.1 by looking at the pretraining loss on Yoruba, the lowest-resource language in our pretraining corpus. Figures 2 and 3 show that UniMax prevents overfitting.
>
> Re: grouping languages — For TyDi-QA, our results are ordered by data availability (see Figure 5), with the lowest-resources languages being Bengali, Telegu and Swahili. We’ve **adjusted the text in §5.2** to highlight the language ordering and discuss low-resource languages in more detail. We’ve also **added two sub-figures to Figure 4** illustrating trends for lower- and higher-resource languages. For WMT, each sub-task involves two languages, so there are many possible options for grouping. We’ve **added breakdowns by English and non-English source and target** in Appendix F. We welcome any additional suggested grouping of pairings.
>
> Re: technical issue with Algorithm 1 — Thanks, we’ve **corrected this notation issue**.

---

### Official Review · Reviewer_ZdPt · 2022-10-25

**Confidence:** 5
**Correctness:** 3
**Technical Novelty And Significance:** 3
**Empirical Novelty And Significance:** 3
**Recommendation:** 6

**Clarity, Quality, Novelty And Reproducibility:**

The paper is very well written, and clearly links its main hypotheses with the actual experiments, providing clear and easy-to-grasp motivation, with clear contributions as well.
- There are some minor clarifications that should be provided in the revised version. In Section 6, it is unclear whether the mT5 model used in comparisons in Table 3 is mT5 from the previous work (i.e., pretrained on the old/uncleaned mC4 corpus) or whether it is the mT5 model pretrained by the authors on the cleaned mC4 corpus).
- Ideally, the paper should also isolate the impact of cleaning the mC4 corpus by: (a) pretraining the full-budget mT5 on the cleaned corpus and comparing it with the old mT5; (b) also running their Unimax pretraining (in the 'reduced-budget' setting with the old mC4 corpus to isolate how much UniMax actually depends on having the cleaned corpus).

Novelty and Originality:
- Tackling the language sampling strategy for multilingual Transformer-based models is not a novel idea, and it has been tackled in some prior work (e.g., see this previous work: https://aclanthology.org/2021.findings-acl.106.pdf). However, doing experiments on pretraining has been limited, mostly due to computational constraints (as the ability to perform extensive comparative empirical analyses is limited when the core focus is put on pretraining). However, the authors did find a good balance between what can versus cannot be done (given the computational limitations), and the paper does a convincing-enough job in supporting the proposed UniMax strategy (which is quite simple in its core). As such, the paper might raise more awareness on improving language sampling in future work as well.

Reproducibility: the paper provides a large number of additional results and design choices in the main paper as well as in the appendix, so I don't have any concerns when it comes to replicating the main experiments from the paper.

**Strength And Weaknesses:**

Strengths:
S1. The proposed strategy is very easy to understand and to implement, it is motivated well (from the high-level hypotheses all the way to empirically showing why it should work, cf., Figure 1, and showing that it works in the actual experiments, cf., Figure 4 and Table 3).

S2. The work is well motivated and very well written - the reader can really follow the main motivation and the authors' line of thinking, which is supported by empirical evidence and insightful analyses throughout the paper.

S3. The proposed strategy holds promise to improve all currently available multilingual LLMs, given sufficient budget for computation - hopefully, this will inspire retraining of most of the current models.

S4. The side contribution (the cleaned mC4 corpus) might be useful for other researchers as well.

Weaknesses:

W1. The choice of tasks is okay, but could be improved - I feel that more emphasis in evaluation should be given to low-resource languages, e.g., perhaps evaluating on the NLI task using the AmericasNLI dataset?

W2. This is not a major weakness as I am fully aware of computational restrictions - the paper focuses on a single model in this paper (mT5) so without providing any extra empirical evidence, it is not guaranteed that UniMax will work with other multilingual Transformers such as XLM-R or X-MOD. It probably will work, but this has to be empirically checked.
- Related to the X-MOD work (Pfeiffer et al., NAACL-2022) - I wonder if the UniMax strategy is equally important for that pretraining approach, where the curse of multilinguality is mitigated by modularising pretraining - is it then equally important to do capping of low-resource languages if each language is associated with its dedicated parameters? The authors should at least discuss this paper as part of related work.

W3. The paper should more clearly isolate the impact of cleaning mC4 versus the impact of the actual proposed sampling strategy.

**Summary Of The Paper:**

This paper presents a simple yet effective strategy to better balance different languages when pretraining a massively multilingual language model (such as mBERT, mT5, XLM-R, or XLM-E). The idea is quite straightforward but nicely motivated and convincingly executed: instead of the typically used temperature-based sampling, the authors propose the UniMax strategy which avoids or mitigates overfitting to low-resource languages as it caps the number of repeats in the skewed distribution of the languages. This capping is done by distributing a predefined character budget as uniformly as possible without using more than N epochs per language (where N is the hyper-parameter).

Using mT5-style pretraining as the case study, the paper then shows the benefits of the UniMax strategy over the standard temperature-based sampling at pretraining, focusing on two multilingual tasks: NMT (on WMT21) and QA on TyDi QA. The empirical findings support the proposed strategy, and the authors offer a series of additional experiments and ablations related to critical strategy choices (e.g., the value for N). They also demonstrate that UniMax works with different computational budgets and yields performance benefits in 'full-budget' as well as 'reduced-budget' pretraining regimes.

As clean language input is pivotal for the strategy to work well, as a side contribution, the authors also clean and filter the original mC4 corpus, yielding higher-quality pretraining data.

**Summary Of The Review:**

 This is a solid work with solid motivation and convincing experiments (also in light of computational demands required for this type of work). The idea is not highly original and not highly exciting, but the final deliverables of the work might have some impact in the field of multilingual representation learning.

The coverage of related work is also really good, showing that the authors are knowledgeable in this field and have done a good job in systematising knowledge from prior work.

I am open to adjusting my score if the authors provide additional information related to some of my questions above.

---

> ### Author Response · Authors · 2022-11-19
> **Response to Reviewer ZdPt**
>
> Thank you for the detailed review.
>
> Re: choice of tasks — We agree it would be useful to evaluate on more tasks, especially those with more emphasis on low-resource languages. We’ve **added several more “standard” tasks** (XNLI zero-shot, XNLI translate-train, XQuAD, PAWS-X) in §6 and Appendix D. However these skew toward higher-resource languages. We’ve also **added more detailed breakdowns by language grouping** where possible (e.g. Figure 4) to help see the effects on low-resource languages.
>
> Re: AmericasNLI — Thanks for the suggestion, we weren’t previously aware of this dataset. This looks like a very useful benchmark. Unfortunately, none of the languages are included in our 101-language pretraining corpus, so it would be hard to interpret changes in performance as we vary the sampling strategy. We hope to return to this with a more diverse pretraining corpus in the future.
>
> Re: focus on mT5 — To the best of our knowledge, mT5 is the current state-of-the-art unsupervised multilingual model, outperforming mBERT and XLM-R. As such, we felt this choice had the potential to be most impactful for the multilingual research community. Computational limitations kept us from exploring alternative model pretraining options.
>
> Re: X-MOD — We expect that UniMax would still be helpful for such models. Since UniMax caps low-resource languages to avoid repeating examples, practitioners can now re-invest this compute budget into additional training on higher resource languages. Unfortunately, we do not have the computational resources to fully explore this. However we have **added a citation of X-MOD and discussion of the applicability of UniMax to other pretraining paradigms** in the conclusion.
>
> Re: impact of cleaning mC4 — We agree this is an important factor to pull apart. We have **added an ablation in Appendix E by training an additional “full-budget” UniMax model on the original mC4 data** to isolate and measure the separate effects of pretraining data and sampling strategy.

---

### Author Response · Authors · 2022-11-19
**Overall Response**

We thank all four reviewers for their helpful comments and suggestions. We believe we have addressed many of the concerns raised. We ask reviewers to consider adjusting their scores accordingly.

Re: limited evals — All reviewers pointed to the need for further evaluations. We have addressed this in four ways:

1) **Conducted additional experiments (21 fine-tuning runs) on XNLI zero-shot, XNLI translate-train, and XQuAD**, with results in Appendix D.
2) **Added an ablation to factor out the contribution of the mC4 data refresh** by pretraining two UniMax models to 100K steps on the old and refreshed corpus, and fine-tuning on XNLI, with results in Appendix E.
3) **Added results on PAWS-X** to Table 3.
4) **Added more detailed groupings to highlight performance on low-resource languages** for both TyDiQA (see new plots in Figure 4) and WMT21 (see Figure 8).

Re: novelty — We view the relative simplicity of our method as a strength, as we believe simple but effective methods have the best chance of being widely adopted. Despite this simplicity, **previous work has not explored epoch-based caps** on training mixture components for language models. We also wish to emphasize our **empirical novelty**—to our knowledge, this is the first study to provide a detailed exploration of the **interaction between language balancing and model scale**. Finally, we believe our findings and accompanying code, data and checkpoints will be directly useful to the NLP community.

Updates to manuscript:
1) Added discussion of XNLI, XQuAD, MLQA, PAWS-X to §4.3.
2) Added evaluations on XNLI (zero-shot and translate-train) and XQuAD in Appendix D.
3) Added an ablation comparing the original vs. refreshed mC4 pretraining corpus in Appendix E.
4) Added PAWS-X evaluation results to Table 3.
5) Added two sub-figures to Figure 4 illustrating trends for lower- and higher-resource languages on TyDi-QA.
6) Added breakdowns on WMT21 task by English vs. non-English source and target in Appendix F.
7) Adjusted text in §5.2 to highlight language grouping and results in low-resource languages.
8) Added to the conclusion discussion of the applicability of UniMax to other pretraining paradigms (encoder-only, decoder-only, using parallel data, X-MOD).
9) Corrected notation issue in Algorithm 1.

Please see our responses to individual reviewers for more detailed discussion around specific comments.

---

### Decision · Program_Chairs · 2023-01-20

**Decision:**

Accept: poster

**Justification For Why Not Higher Score:**

More evaluation could be helpful to make the paper more convincing, for example, with more models and datasets.

**Justification For Why Not Lower Score:**

This paper addresses an important problem for the training of multi-linguil PLMs.  The proposed methods is effective.
The authors' response well addressed most of the concerns raised by the reviewers.

**Metareview: Summary, Strengths And Weaknesses:**

Summary:

This paper proposes a simple yet effective strategy for language data sampling for training a large multilingual pre-trained language model, to avoid oversampling low-resource languages, which often happens with the temperature-based strategy.  Experiments on mT5-style models and two multilingual tasks show benefits of the proposed method in various experiment settings.  A filtered subset of mC4 corpus is provided as a higher-quality praining data for researchers.


Strength:

The proposed strategy is very easy to understand and to implement.
The work is well motivated and very well written.
The approach is straightforward, and the results outperform the other sampling ratio baselines.
A new datasets for multilingual LM training will be released.


Weaknesses:

More evaluation could be helpful to make the paper more convincing, for example, with more models and datasets.


**Note From Pc:**

if the above contains the word "oral" or "spotlight" please see: "oral" presentation means -> notable-top-5% and "spotlight" means -> notable-top-25%. As stated in our emails, we are disassociating presentation type from AC recommendations

**Summary Of Ac-Reviewer Meeting:**

NA